# Relative Importance of Determinants of Changes in Eating Behavior during the Transition to Parenthood: Priorities for Future Research and Interventions

**DOI:** 10.3390/nu13072429

**Published:** 2021-07-15

**Authors:** Vickà Versele, Phaedra Debekker, F. Marijn Stok, Dirk Aerenhouts, Peter Clarys, Benedicte Deforche, Eva D’Hondt, Roland Devlieger, Annick Bogaerts, Tom Deliens

**Affiliations:** 1Department of Movement and Sport Sciences, Faculty of Physical Education and Physiotherapy, Vrije Universiteit Brussel, Pleinlaan 2, 1050 Brussels, Belgium; phaedra.debekker@vub.be (P.D.); dirk.aerenhouts@vub.be (D.A.); peter.clarys@vub.be (P.C.); benedicte.deforche@ugent.be (B.D.); eva.dhondt@vub.be (E.D.); tom.deliens@vub.be (T.D.); 2Department of Development and Regeneration, Faculty of Medicine, KU Leuven, Herestraat 49, 3000 Leuven, Belgium; roland.devlieger@uzleven.be (R.D.); annick.bogaerts@kuleuven.be (A.B.); 3Department of Interdisciplinary Social Science, Utrecht University, Heidelberglaan 1, 3584 CS Utrecht, The Netherlands; f.m.stok@uu.nl; 4Department of Public Health and Primary Care, Faculty of Medicine and Health Science, Ghent University, C. Heymanslaan 10, 9000 Ghent, Belgium; 5Obstetrics and Gynaecology, University Hospitals KU Leuven, Herestraat 49, 3000 Leuven, Belgium; 6Faculty of Medicine and Health Sciences, Centre for Research and Innovation in Care (CRIC), University of Antwerp, 2000 Antwerp, Belgium; 7Faculty of Health, University of Plymouth, Plymouth, Devon PL4 8AA, UK

**Keywords:** first-time parents, pregnancy, postpartum, nutrition, diet, research and intervention priorities

## Abstract

Background: Healthy eating behavior throughout pregnancy and postpartum is important. This study aimed to investigate the perceived sex-specific importance of determinants of changes in eating behavior during pregnancy and postpartum. Methods: Fifty-four determinants were rated by first-time parents (*n* = 179) on their impact. Experts (*n* = 31) rated the determinants in terms of their modifiability, relationship strength, and population-level effect from which a “priority for research”-score was calculated. Results: During pregnancy, the three highest rated determinants by women were “health concerns”, “physiological changes”, and “fatigue”. Men perceived “health concerns”, “health consciousness”, and “influence of the pregnant partner” as important. Postpartum, the three highest rated determinants by women were “adaptation to rhythm of baby”, “baby becomes priority”, and “practical constraints because of the baby”. Men perceived “adaptation to rhythm of baby”, “fatigue”. and “(lack of) anticipation” as important. According to the experts, “professional influence”, “food knowledge”, and “home food availability” received high priority scores for both sexes and during both periods. Conclusions: Priority for research and interventions should go towards tailored family-based approaches focusing on food education in a broad sense taking into account aspects such as health consciousness, self-efficacy skills, and the social and home food environment while being supported by healthcare professionals.

## 1. Introduction

The transition to parenthood is an important period in life for both women and men during which changes in eating behavior may occur [1]. Improving or maintaining healthy eating behavior during the course of pregnancy and in the postpartum period is of utmost importance for both the mother and the father. Healthy eating behavior reduces the risk for excessive gestational weight gain (GWG) and facilitates controlling weight-related outcomes in women during the postpartum period [2,3,4,5]. Besides, also fathers are found to be at risk for unfavorable weight changes during the pregnancy and postpartum period [6,7]. Weight changes during the pregnancy and the early postpartum period might contribute to long-term weight gain and overweight/obesity later in life [8]. Hence, healthy eating behavior is an important aspect to prevent undesirable weight gain in both parents.

Improving or maintaining healthy dietary behavior during pregnancy and postpartum impacts the next generation as well. In the short term, a mother’s eating behavior is linked with birth outcomes (e.g., small- or large-for-gestational age, pre-term birth). In the long term, the healthy eating behavior of parents is associated with the offspring’s health (e.g., reduced risk of obesity, cardiovascular diseases), as well as with their eating behavior when parents become a role model for their child/children [9,10]. The family environment is linked with children’s food intake and weight status, with children being more likely to become overweight or obese themselves if both of their parents are overweight or obese. This highlights the need to prevent undesirable changes in eating behavior and excessive weight gain among parents [11,12].

Eating behavior and nutritional intake during pregnancy and in the postpartum period are driven by food choices people make during this life-changing period. Understanding why and how people make certain (un)healthy choices is needed to be able to direct people towards better food choices. Both women and men have been shown to change their diet during pregnancy and in the postpartum period [13,14,15,16,17]. These changes are driven by the interplay of numerous determinants at the individual (e.g., cravings, fatigue, lack of knowledge or skills, and time constraints), interpersonal (e.g., professional influence, social pressure to eat, and dietary intake of the baby), and environmental level (e.g., home food availability) [14,18,19,20,21].

Determinants of changes in eating behavior on an individual (subdivided into biological, psychological, and situational sublevels), interpersonal, and environmental level (subdivided into micro- and meso-/macro-sublevels) have been defined in a qualitative focus group study in a sample of expecting and first-time parents in Belgium [14]. Determinants influencing changes in eating behavior are modifiable to a certain extent, and food choices and eating behavior show great diversity across individuals and populations [22]. However, in order to make statements about the (sex-specific) importance of the determinants and to know which determinants should be prioritized when targeting women, men, or the couple as a whole, a quantitative assessment of the determinants of changes in eating behavior during the transition to parenthood, identified in the previous qualitative explorative study, is required [14]. Diverse frameworks with different determinants of nutrition and eating behavior have been designed, to systematize the coherence of the determinants [22]. The Determinants Of Nutrition and Eating (DONE) framework, developed by Stok et al., is an example of such an interdisciplinary framework to explain eating behavior in a general population [22]. By scoring each determinant of the framework on modifiability, relationship strength, and population-level effect, a priority for research can be calculated [22]. The DONE framework thus allows the identification of determinants of interest that can be used in targeted interventions. To date, no quantitative assessment of the determinants of changes in eating behavior, aimed at defining priority areas to target when developing interventions in expecting and first-time mothers and fathers, exists.

A further and better understanding of the importance of the determinants influencing both women’s and men’s changes in eating behavior during the transition to parenthood, taking into account the difference in importance of each determinant between women and men, is needed. Therefore, the present study aimed at determining the sex-specific importance of determinants explaining changes in eating behavior during the transition to parenthood by rating these determinants by both the target population itself and a group of (inter)national experts.

## 2. Materials and Methods

For the purpose of this study, a quantitative analysis aiming to investigate the importance of determinants of changes in eating behavior during pregnancy and postpartum was performed. This was done based on qualitative data obtained through previously conducted focus group discussions with both women and men [14].

### 2.1. Study Flow

Focus group discussions were used to investigate determinants of changes in eating behavior during pregnancy and in the postpartum period (up to 1 year) in first-time mothers and fathers [14]. Two frameworks were developed based on determinants of changes in eating behavior during the transition to parenthood: one related to the pregnancy period and one related to the postpartum period. Based on these frameworks, online questionnaires were developed enabling the rating and evaluation of the importance of each determinant by a sample of first-time parents, as well as by academic experts (see Figure 1).

### 2.2. Step 1: Development of Determinant Frameworks through Focus Group Discussions

Using a semi-structured question guide during 13 focus group discussions, 74 expecting and first-time parents were asked to identify changes in their eating behavior during pregnancy and in the postpartum period. Data analysis was performed through an inductive thematic approach, resulting in a list of 54 determinants of changes in eating behavior (33 determinants during pregnancy, 21 determinants postpartum). The determinants were systematically categorized into individual (three sublevels: biological, psychological and situational), interpersonal, and environmental (two sublevels: micro and meso/macro) levels of influence. A more detailed description and overview of the focus group study’s methodology and findings can be found elsewhere [14].

### 2.3. Step 2: Evaluation of Determinants through Online Questionnaires

The perceived importance of the obtained determinants was evaluated in two different ways, namely by a sample of first-time parents and by a group of (inter)national academic experts within one or more of the following fields of expertise: (1) maternal health (i.e., gynecology and obstetrics, maternal and child health, maternal obesity, gestational diabetes); (2) nutrition/physical activity (i.e., human nutrition and dietetics, physical activity and movement sciences); (3) public health (i.e., social and behavioral sciences, public health, epidemiology and health policy).

#### 2.3.1. Recruitment and Data Collection within the Sample of First-Time Parents

Participants from two previous studies (i.e., participants of the focus group discussions [14] and participants from the longitudinal TRANSPARENTS study [23]) were contacted through email and asked to participate in the present study. A total of 340 first-time parents were contacted (of which 173 were women and 167 men).

The study sample of first-time parents received an email explaining the aim of the study and a link to the online questionnaire. Before being able to start the questionnaire, all participants had to give informed consent on an individual basis. The study protocol was approved by the Medical Ethics Committee of the university hospital (Vrije Universiteit Brussel, Brussels, Belgium). The study sample was randomly (through a random selection formula in Excel) divided into two groups of 170 participants. The first group (89 women and 81 men) received the questionnaire covering the 33 determinants of changes in eating behavior during pregnancy, whereas the second group (86 women and 84 men) received the questionnaire covering the 21 determinants of changes in eating behavior within the first year postpartum. In order to eliminate test-order effects, the determinants within each questionnaire were randomly (through a randomized survey flow setting in the questionnaire software, Qualtrics) sequenced. Participants were given two weeks to complete the questionnaire; a reminder was sent by email after one week. Experts were given one month to complete the questionnaire; a reminder was sent twice, the first one week after the initial email and the second after two weeks.

#### 2.3.2. Recruitment and Data Collection within the Sample of (Inter) National Academic Experts

The group of (inter)national experts was selected from the personal network of the research team involved and by screening reference lists of relevant literature on the topic of eating behavior during pregnancy and postpartum. Consequently, a list of 144 experts was created, all of whom were contacted through email. In an attempt to reach more experts, the experts were asked to forward the questionnaire to colleagues with similar expertise in their research group.

Similar to the data collection for the sample of first-time parents, the (inter)national experts received an email with the explanation about the aim of the study, why they were selected as an expert, and a link to the online questionnaire. All experts had to give consent to participate before being able to start the questionnaire. Determinants were randomized (through a randomized survey flow setting in the questionnaire software, Qualtrics) individually within each period (i.e., during pregnancy and within the first year postpartum) to avoid test-order effects.

#### 2.3.3. Questionnaires

Questionnaires were developed in Dutch (for parents) and English (for experts) and split up based on the period (pregnancy vs. postpartum) for which the corresponding determinants were listed (*n* = 33 vs. *n* = 21, respectively). The first part of the questionnaire for the parents consisted of questions related to sociodemographic characteristics (i.e., age, educational level), self-reported body weight (in kg) and height (in cm), the birthday of the first child and whether she were pregnant for the second time. Experts were first questioned about their sex, age, area of expertise, and years of research experience.

The second part of the questionnaires concerned the questions related to the rating of the determinants. An explanation of each determinant within each (sub)level was foreseen and, where possible, accompanied by a specific example. For instance, the determinant “Food knowledge” was accompanied by the following explanation: *“Facts, information, and acquaintance acquired through experience or education about food. Lack of information about what can be eaten and what not during pregnancy (of the partner)”*. An overview of all determinants of changes in eating behavior during the pregnancy and postpartum period together with the explanation and examples can be found in Appendix A.

The group of first-time parents were asked to allocate a score from 0 (i.e., no impact at all on changes in their eating behavior) to 10 (i.e., very high impact on changes in their eating behavior) for each determinant. Determinants were scored by both women and men, except for “physiological changes” (i.e., only applicable for women) from the biological sublevel at the individual level.

In accordance with the DONE scoring protocol [22], the sample of (inter)national experts was asked to rate each determinant for women and men, separately, on three dimensions: (1) modifiability (score of 1–3, i.e., low, medium, or high); (2) strength of the relationship between the determinant and changes in eating behavior (score of 1–2, i.e., correlational or causal); and (3) population-level effect (score of 1–3, i.e., low, medium, or high). Each determinant had to be rated separately for women and men, except for “physiological changes” (i.e., only applicable for women) and “influence of female/male partner” (i.e., only applicable for the opposite sex). When the expert thought a particular determinant was not applicable (N.A.) for women or men, “N.A.” could be selected. To quantitatively take into account that a determinant was not applicable in explaining changes in eating behavior, a zero was allocated to the three dimensions altogether when a determinant was indicated as “N.A.”. Mean and SDs were calculated for each of the three dimensions. To define areas of research and intervention focus, a “priority for research”-score per determinant was calculated following the example of previous studies that have investigated factors influencing eating behavior [22,24,25]. The following formula to adjust for the different levels of scoring options between the three categories was used: (mean score on modifiability/3 + mean score on relationship strength/2 + mean score on population-level effect/3), with higher scores reflecting higher priority for research. At the end of each list of determinants (pregnancy vs. postpartum), two closing questions were asked: *“In your opinion, are there determinants missing in this list?”* and *“Would you change the name of any of the determinants in this list?”*, in accordance with the method used by Stok et al. [22].

### 2.4. Data Analysis

The questionnaires were developed, distributed, and managed by Qualtrics Version XM (Qualtrics, Provo, UT, USA). SPSS Statistics Version 26 (IBM Corp, Armonk, NY, USA) was used to perform the statistical analyses. Sample characteristics and the mean score ± SD of each determinant were calculated for women and men separately. Independent samples *t*-tests were used to examine sex differences between the mean scores of the determinants of changes in eating behavior of first-time mothers and fathers. Significance was considered at an alpha level of 0.05. The graphs of the rated determinants were created in Microsoft Excel 16.16.22 (Microsoft Corporation, Albany, NY, USA) and the updated frameworks based on the expert ratings was visualized using Tableau Software Version 9.1. (Tableau international, Seattle, WA, USA).

## 3. Results

The top five determinants perceived as most important by the first-time parents and experts are mentioned in the text. The perceived importance of all determinants can be found in Figure 2, Figure 3, Figure 4, Figure 5, Figure 6 and Figure 7 and Appendix A.

### 3.1. Rating of the Determinants by First-Time Parents

A total of 179 out of 340 invited first-time parents (response rate of 52.6%, i.e., 60.7% of 173 invited women and 44.3% of 167 invited men) completed the questionnaire. Of the 170 first-time parents who received the questionnaire with determinants of changes in eating behavior during pregnancy, 98 participants started the questionnaire. As 4 participants only completed the first part with sociodemographic characteristics, the final sample consisted of 94 participants with an age ranging between 24 and 40 years (women: 24–37 years, men: 28–40 years). Of the 170 first-time parents who received the questionnaire with determinants of changes in eating behavior within the first year postpartum, 96 participants started the questionnaire. As 5 participants only gave their informed consent and 6 only completed the first part with sociodemographic characteristics, the final sample consisted of 85 participants with an age ranging between 25 and 40 years (women: 25–39 years, men: 28–40 years). The characteristics of the study sample are shown in Table 1.

An overview of determinants of changes in eating behavior during pregnancy is shown in Figure 2. A detailed overview of the mean scores ± SD and the t- and *p*-values of the sex differences between ratings of each determinant are presented in Appendix A.

The five most important determinants perceived to explain changes in eating behavior during pregnancy according to women comprised “health concerns”, “physiological changes”, “fatigue”, “(perceived) food safety”, and “discomfort”. For expecting men, the five most important determinants perceived to explain changes in their eating behavior were: “health consciousness”, “health concerns”, “influence of female partner”, “weight control”, and “home food availability”.

At the biological sublevel, women perceived all determinants as more important than men (all *p* ≤ 0.001). At the psychological sublevel, the determinants “food knowledge” (*p* = 0.636), “habits” (*p* = 0.116), “health consciousness” (*p* = 0.313), “mood and emotions” (*p* = 105), “self-control” (*p* = 0.053), and “self-efficacy” (*p* = 0.107) were perceived as being equally important to both women and men expecting their first child. “(Perceived) food safety” (*p* < 0.001), “health concerns” (*p* < 0.001), “worries and concerns” (*p* = 0.001), “self-licensing” (*p* = 0.029), “anticipation” (*p* < 0.001), “eating regulation” (*p* = 0.005), and “weight control” (*p* = 0.027) were scored higher by women than men. At the situational sublevel, the determinants “time constraints” (*p* = 0.177) and “other priorities” (*p* = 0.566) were not scored differently by both women and men, whereas “effort and convenience” (*p* = 0.002) received a higher score by women than men. At the interpersonal level, women and men perceived a comparable influence on their eating behavior through the “influence of the partner” (*p* = 0.588) and “social pressure to eat” (*p* = 0.825). Women scored “professional influence” (*p* = 0.001) and “sensitivity to others’ opinion” (*p* = 0.018) higher. Finally, at the environmental level, “home food availability” (*p* = 0.613) was perceived to be equally important for both expecting parents, whilst “environment food availability” (*p* = 0.030) was perceived more important by women.

All perceived determinants of changes in eating behavior during the postpartum period are shown in Figure 3. Mean scores ± SD of rated determinants and the t- and *p*-values of the sex differences are presented in Appendix A.

The five most important determinants perceived to explain changes in eating behavior during the postpartum period for women were “adaptation to rhythm of baby”, “baby becomes priority”, “practical constraints because of baby”, “baby needs attention”, and “dietary intake of baby”. For men, “fatigue” and “anticipation” were the highest rated determinants perceived to explain changes in eating behavior postpartum, whilst “baby needs attention”, “baby becomes priority”, and “planning” also received a high score.

At the individual level, determinants within the biological and situational sublevel were perceived equally important for first-time mothers and fathers (all *p*-values between 0.050 and 0.313). “Self-licensing” (*p* = 0.036), “self-control” (*p* = 0.038), and “weight control” (*p* = 0.027) were perceived as more important for first-time mothers than fathers. At the psychological sublevel, the determinants “food knowledge” (*p* = 0.681), “habits” (*p* = 0.509), “anticipation” (*p* = 0698), and “planning” (*p* = 0.237) were perceived to have a comparable influence for both partners. Except for “professional influence” (*p* = 0.083) and “role model” (*p* = 0.102), being perceived as equally important for women and men, all determinants at the interpersonal level (i.e., all determinants related to the influence of the baby) were scored higher by women than men (all *p*-values between 0.003 and 0.048). No sex differences in importance were found at the environmental level (i.e., “home food availability” (*p* = 0.138)).

### 3.2. Rating of the Determinants by (Inter)National Experts

A sample of 144 experts was contacted, of which 45 experts started the questionnaire. As 14 experts only completed the first part with socio-demographic characteristics, the final sample consisted of 31 experts (response rate of 22.2%) who rated (part of) the determinants. Because some experts did not complete the entire questionnaire and due to the random sequenced order of the determinants within the questionnaire, each determinant was rated on three dimensions by 24 to 30 experts. Characteristics of the experts are shown in Table 2.

The mean scores of the determinants explaining changes in women’s and men’s eating behavior during pregnancy and postpartum are presented in Figure 4, Figure 5, Figure 6 and Figure 7. These figures simultaneously depict the scores for each determinant on the three rating dimensions: modifiability (y-axis), relationship strength (circle size), and population-level effect (x-axis). Determinants in the top-right corner with larger sized circles received the highest expert scores on modifiability, relationship strength, and population-level effect and thus can be considered as priorities for further research and policy. The color of the circles depicts the (sub)level of the determinants: blue for the individual level (darkest blue: situational sublevel, medium blue: psychological sublevel, light blue: biological sublevel), orange for the interpersonal level, and green for the environmental level (darkest green: micro-sublevel, light green: meso-/macro-sublevel).

The five determinants receiving the highest “priority for research”- score for women during pregnancy were “professional influence” (2.43), “food knowledge” (2.41), “health concerns” (2.39), “health consciousness” (2.31), and “habits” (2.30) (see Figure 4). The highest “priority for research”-scores for men during the pregnancy of their partner were found for “food knowledge” (2.13), “health consciousness” (2.09), “habits” (2.07), “self-efficacy” “(1.96), and “home food availability” (2.10) (see Figure 5).

The five determinants receiving the highest “priority for research”-score for women during the postpartum period were, “home food availability” (2.35), “professional influence” (2.31), “food knowledge” (2.29), “weight control” (2.27), and “role model” (2.18) (see Figure 6). The determinants with the highest “priority for research”-score for men during the postpartum period were “home food availability” (2.34), “food knowledge” (2.21), “professional influence” (2.07), “role model” (2.03) “planning” (2.01), and “dietary intake of the baby” (2.01) (see Figure 7).

A detailed overview of the mean scores ± SD on modifiability, relationship strength, and population-level effect, as well as the “priority for research”-score of all determinants among both women and men during the pregnancy, as well as the postpartum period can be found in Appendix A.

Suggestions were added by the experts regarding the naming of the determinants or concerning determinants that could have been missed during the focus group discussions (e.g., due to the homogenous study sample). These suggestions were discussed by the authors in order to determine their usefulness. From the suggested “missed determinants”, most could be categorized under an already-existing determinant (e.g., “limited appetite”, which would be classified within “hunger and satiety” (biological sublevel), and “influence of parents”, which is covered under “social influence” (interpersonal level)). Some determinants seemed valid suggestions (e.g., “risk perception for Gestational Diabetes Mellitus (GDM)”), but only for subgroups not represented during the focus group discussions (e.g., pregnant women with GDM) that preceded the present study [14].

## 4. Discussion

The present study investigated the relative importance of predefined determinants of changes in eating behavior during pregnancy and postpartum in first-time parents and to what extent this was different for women compared to men. To the best of our knowledge, such a quantitative assessment of qualitatively defined determinants perceived to explain changes in eating behavior during the transition to parenthood has never been done before. In summary, both parents and experts perceived health-related determinants as important in explaining changes in eating behavior during pregnancy. The influence of the baby was perceived as the most important determinant by the parents during the postpartum period. Experts considered food knowledge, professional influence, and the home food environment as important to focus on for both (expecting) parents and during both pregnancy and postpartum period.

Health-related psychological determinants such as “health consciousness”, “health concerns”, “(perceived) food safety”, and “weight control” were perceived to be important for both parents and experts in explaining changes in eating behavior among expecting and first-time parents. It has been shown that becoming a father motivates men to eat more nutritiously, exercise more often, and drink less alcohol [15], which can be linked with their health consciousness. Furthermore, women are motivated to eat healthier when becoming a mother [26]. However, due to the growing fetus, pregnant women are often concerned about the impact of their diet during pregnancy [27], which could explain the high scores on “health concerns” and “(perceived) food safety” in first-time mothers. This might result in reducing foods that may harm pregnancy or the mother’s health, rather than an increase in nutritious foods providing nutrients required during pregnancy [18]. Many women also experience challenges in preventing excessive GWG, while postpartum weight retention is also an often described problem [28]. Fathers seem to be confronted with weight gain during pregnancy of their partner and as a result of fatherhood [6,7,14]. In addition to the abovementioned determinants, fathers perceived “planning” as important, and experts considered “habits” and “self-efficacy” as important psychological determinants to target. Furthermore, other research has shown that determinants such as planning and habits influence behavior, while self-efficacy is described in relation to health behavior change [29,30]. These determinants seem to be important to target when developing effective interventions [31]. Unhealthy habits may attenuate the impact of intervention strategies [30], and maintaining existing healthy pre-pregnancy habits/behaviors is thus as important as obtaining new health-related habits/behaviors during pregnancy and postpartum. This could be done through the use of self-control strategies (e.g., self-monitoring, planning) and is recommended to make interventions more effective [30].

Many of the aforementioned determinants may be associated with food knowledge. Even though not in the top five, expecting fathers perceived “food knowledge” as important in explaining changes in their eating behavior, which is in contrast to expecting mothers and first-time parents. This is remarkable as one might expect mothers-to-be to be aware of the importance of food knowledge, especially since “(perceived) food safety” was mentioned as a determinant of changes in their eating behavior [14]. The experts on the other hand considered “food knowledge” as an important determinant in terms of modifiability, relationship strength, and population-level effect for both parents during both periods. However, a focus solely on improving food knowledge does not seem to be effective in motivating people towards behavior change [31]. Education, which is needed to increase food knowledge, is key, but should be used alongside other types of interventions (e.g., behavioral change interventions making use of goal setting and/or self-monitoring [31,32]) to increase intervention efficacy [33,34,35]. Furthermore, “professional influence” was rated as an important determinant by the experts. For expecting women, this was not in the top five of determinants, but still among the seven highest rated determinants. Expecting fathers and parents in the first year postpartum did not perceive the influence of professionals as important. Research has shown that expecting parents lack knowledge and experience insufficient (professional) support regarding nutrition and eating behavior during and after pregnancy [14,36,37]. Hence, (expecting) parents—and especially expecting fathers—should be supported and taught about food knowledge in a broad sense where aspects such as health consciousness, food safety, healthy eating habits, and self-efficacy skills are triggered. Pregnancy and postpartum weight management interventions that are based on individualized nutritional support and improving self-monitoring skills appear to be the most successful [28,38,39]. Therefore, healthcare providers could play an important role in educating and supporting expecting and first-time parents in order to improve or maintain healthy eating behavior, even when encountering unmodifiable pregnancy- (e.g., “physiological changes”, “discomfort”) and postpartum- (e.g., “fatigue”) related issues. How to cope with these barriers has to be taken into account when supporting expecting and first-time parents.

Our results indeed showed that biological determinants such as “discomfort”, “physiological changes”, and “fatigue” were perceived to be important in explaining changes in eating behavior in pregnant women. Others have described the impact of discomfort and physiological changes on eating behavior during pregnancy, but not on changes in eating behavior, highlighting once again the uniqueness of the present study [40,41]. As these biological determinants are very individually determined and have limited modifiability, it is no surprise that these were not considered as priorities by the experts. Postpartum, “fatigue” remains an important determinant, especially for first-time fathers, along with the determinants related to the baby (e.g., “adaptation to rhythm of baby”), which were highlighted by both parents as key determinants. It is known that the postpartum period is a challenging period for young parents. For women, physical recovery after childbirth, fatigue due to sleep deficiency, and emotional fatigue are some of the many postpartum challenges [42,43]. The impact of the child appears to be very important to women, which could be due to the fact that women naturally feel more involved in the care of a child [42]. Men, in their turn, struggle with new demands, the conflict of several aspects of value in their life, and their new role as protector and provider of the family [42]. For both parents, taking care and meeting the needs of a baby take responsibilities and much time and effort. Spending time with the child might become a priority over doing groceries or providing and devoting time to preparing healthy meals [14]. On the other hand, (upcoming) parenthood might also be a window of opportunity during which first-time parents may be more motivated to improve their lifestyles [6,7,44]. Nonetheless, these physiological, psychological, and social challenges during the postpartum period make it difficult to shape healthy behaviors [28]. The new barriers and demands of parenthood highlight the need for interventions with a flexible and personalized approach. The new family demands and parents’ self-management skills have thus to be taken into account, making it possible for first-time parents to maintain or obtain a healthy eating behavior.

The social and home environment were also considered important. For example, parents attributed high scores to the influence of their partner on their personal eating behavior; for fathers, this was even perceived as the third most important determinant. The support of expecting and first-time parents by their partner, but also family, friends, and professionals is needed and very important [42]. A strong social network could, for example, help in providing a healthy home food environment during this challenging period by offering a healthy meal or support with groceries. Experts—but also parents—considered “home food availability” as an important determinant to focus on during pregnancy and postpartum. It has been shown that there is a positive association between the exposure and availability of foods (at home) of low-income pregnant women and their daily intake [45]. It is likely that when (expecting) parents are confronted with barriers such as fatigue, they often will reach for what is available at home, especially when extra demands/constraints related to the baby arise during the postpartum period. However, in previously conducted qualitative research, the dietary intake of the baby was mentioned as a motivator to make sure fresh foods such as fruits and vegetables are available at home [14]. Improving home food availability and making healthy nutrition easily available for expecting parents and young families would be a good starting point for further research and interventions, especially in precarious social and economic situations. Furthermore, the importance of parents as a role model was acknowledged by the experts in our study. Other research already demonstrated the role of parents and their responsibility in the eating behavior of their children [46]. Explaining this might be a motivation for parents to eat healthy themselves. The same qualitative study has shown that parents themselves tried to eat a broader variety of vegetables as a result of the vegetables the child is recommended to eat [14]. Attention should thus equally go to the dietary intake of the parents, as well as that of the child. The role of the social and home environment, including the partner, illustrates the importance of family-based interventions. Family-based approaches targeting health-related behavior are shown to be more effective compared to interventions solely targeting one of both expecting parents [47]. As most determinants discussed above and recommended to focus on in future research and intervention development (i.e., “health consciousness”, “self-efficacy”, “habits”, “professional influence”, “influence of the partner”, and “home food availability” during pregnancy; “fatigue”, “professional influence”, and “home food availability” postpartum) are equally important to women and men, the same intervention may target and reach both parents simultaneously.

Although healthcare providers are expected to play an important role during the transition to parenthood, they experience barriers such as time constraints, gaps in their own nutritional knowledge, and a lack of communication skills, preventing them from providing adequate nutritional support and assistance on improving parents’ self-regulation skills [36,48]. Therefore, educating healthcare providers about nutritional knowledge, the needs and barriers that expecting and first-time parents experience, and training their motivational interviewing skills is recommended [49]. A particular emphasis on communication techniques may be important. This must enable healthcare providers to reflect on their own values and beliefs around health and well-being, which is needed in order to properly communicate [48]. Using principles of motivational interviewing may facilitate parents’ behavioral change [32,49,50]. Prior to the development of tailored and feasible interventions, it seems relevant to investigate healthcare providers’ personal (training) needs, as well as opinions and ideas on how they can support first-time parents (to-be). As healthcare providers are perceived to play a crucial role during this transitional phase, they should be involved as key stakeholders (amongst the parents themselves, but also health insurance companies, health policy makers, etc.). Through the involvement of different groups of healthcare providers (e.g., gynecologists, midwives, GPs, dietitians) and other stakeholders involved in pre- and post-partum care, efforts can be made towards embedding developed interventions in the existing healthcare system.

### Strengths and Limitations

A first strength of this study is the inclusion of results from both first-time parents and experts. Inter(national) experts with different expertise related to this field of research were recruited. The diversity of scientific backgrounds ensures an overall rating of the determinants from a multidisciplinary perspective. Determinants were scored by the parents themselves, while simultaneously asking experts to do the same in order to draw up a score depicting priority for research and interventions. The ratings of importance by the first-time parents along with the ratings on modifiability, relationship strength, and population-level effect by the experts allowed identifying key determinants that should be targeted by future interventions, increasing their likelihood of success. Second, this study addresses both the pregnancy and postpartum period, including both women and men together. As there is a limited amount of literature on changes in eating behavior among (expecting) fathers, this study contributes to the knowledge and insights that are needed to develop and tailor family-based interventions.

A first limitation of this study might be the way in which determinants might have been understood. Even though all determinants were explained in the best possible way with a definition and the addition of examples, there may have been some misinterpretations. Second, first-time parents participating in our study completed the questionnaire during their postpartum period, which may have led to recall bias in the scoring of the determinants related to the pregnancy period. Furthermore, a part of the sample was already expecting a second child (14.5%), which may have influenced the scoring of the determinants. Third, the representativeness of the sample of first-time parents and experts might be questioned. Our sample of first-time parents was mainly Caucasian and highly educated, which may limit generalizability to other (sub)populations. Furthermore, a minority of experts had limited years of research experience. We do not believe this to have influenced our results to a great extent, as we relied on the integrity/self-selection of the experts to only forward/complete the questionnaire if suitable. Fourth, the response rate of the experts was rather low (22.2%). A possible explanation is the length of the questionnaire (about 45 min to complete) and the period during which questionnaires were distributed (end of the academic year in Europe). As the experts were asked to forward the questionnaire to other expert colleagues, the exact response rate could not be determined and might even be slightly lower. Given that our aim was to incorporate the opinion of experts from different fields of expertise in order to gain more insight into which determinants are priorities for further research, and not pursuing statistical inference, we do not consider this low(er) response rate to be problematic. Fifth, a test–retest reliability check of the rating performed by the parents was not performed. Although this might have influenced the absolute scores, we do not expect this to have influenced the relative importance of each determinant. In an attempt to standardize responses, the experts were asked to complete the questionnaire without “overthinking”, highlighting that their “first” reaction was important. This was done in analogy with the method used by Stok et al. [22]. Finally, due to the inclusion criteria (i.e., healthy participants) of the focus group study from which the present determinants were identified [14], specific determinants (e.g., risk perception for pregnancy-related diseases such as GDM) might be missing in the frameworks. These frameworks may thus be further expanded and rated for specific subpopulations, such as same-sex parents or pregnant women with underlying conditions such as GDM.

## 5. Conclusions

An overview of important determinants of changes in eating behavior during pregnancy and postpartum was made based on ratings from first-time mothers and fathers, as well as experts in the field of maternal health and/or nutrition/physical activity and/or public health, providing priority scores for each of the determinants. Both parents and experts perceived health-related psychological (e.g., health concerns), social (partner support), and environmental (home food availability) determinants important when explaining changes in eating behavior among expecting and first-time parents. These determinants can be targeted by providing professional support and education, which were considered as priorities to focus on by the experts. Future research and interventions aiming at improving or maintaining expecting and first-time parents’ healthy eating behavior should thus focus on tailored family-based approaches, supported through healthcare professionals and focusing on food education in a broad sense, incorporating behavioral regulation, the aspect of partner support, and the importance of a healthy home food environment.

## Figures and Tables

**Figure 1 nutrients-13-02429-f001:**
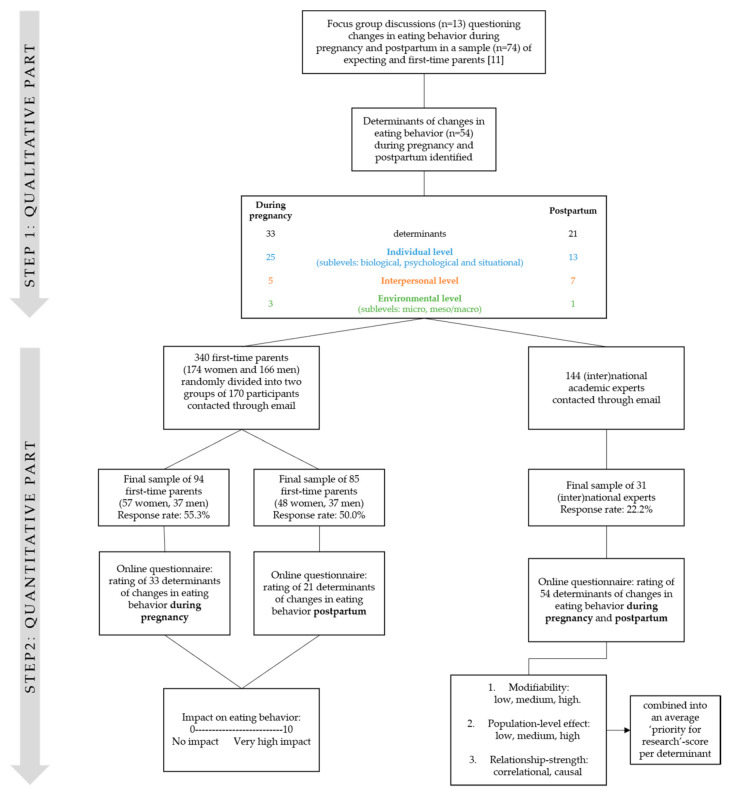
Flowchart of the study methodology and participant recruitment.

**Figure 2 nutrients-13-02429-f002:**
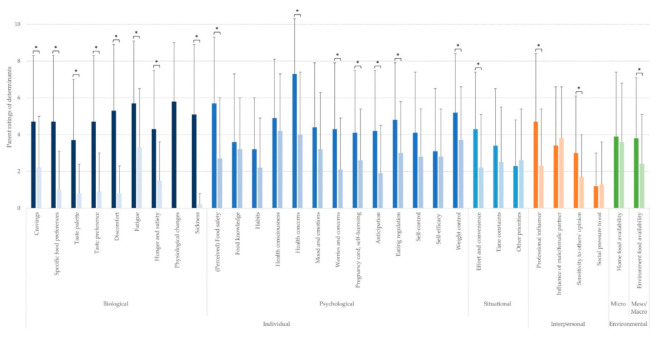
Parents’ rating of determinants of changes in eating behavior (mean ± SD; rated on a scale from 0 to 10) during pregnancy. * Statistically significant sex differences. The colors of the bars depict the (sub)level of the determinants: blue for the individual level (darkest blue: biological sublevel, medium blue: psychological sublevel, lightest blue: situational sublevel), orange for the interpersonal level, and green for the environmental level. The left bar (i.e., darker color) of each determinant depicts the score for pregnant women, and the right bar (i.e., lighter color) of each determinant depicts the score for (expecting) fathers.

**Figure 3 nutrients-13-02429-f003:**
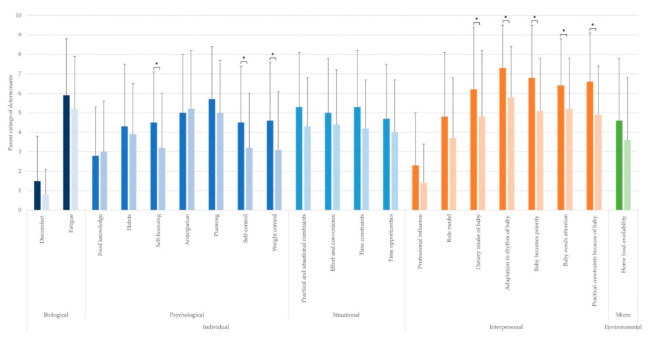
Parents’ rating of determinants of changes in eating behavior (mean ± SD; rated on a scale from 0 to 10) during the postpartum period. * Statistically significant sex differences. The colors of the bars depict the (sub)level of the determinants: blue for the individual level (darkest blue: biological sublevel, medium blue: psychological sublevel, lightest blue: situational sublevel), orange for the interpersonal level, and green for the environmental level. The left bar (i.e., darker color) of each determinant depicts the score for first-time mothers, and the right bar (i.e., lighter color) of each determinant depicts the score for first-time fathers.

**Figure 4 nutrients-13-02429-f004:**
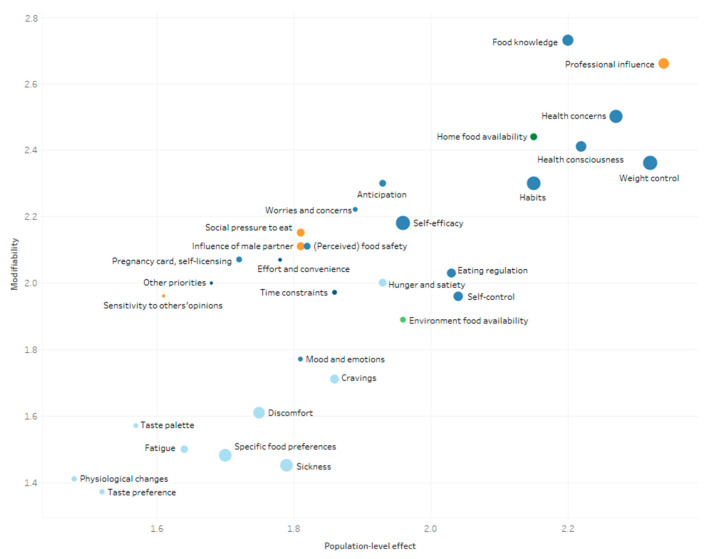
Expert ratings on modifiability, relationship strength, and population-level effect of women’s determinants during pregnancy.

**Figure 5 nutrients-13-02429-f005:**
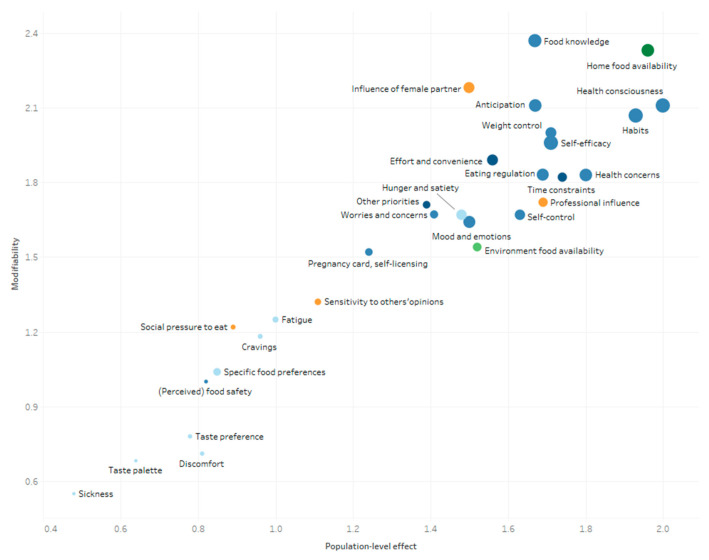
Expert ratings on modifiability, relationship strength, and population-level effect of men’s determinants of changes in eating behavior during the pregnancy of their partner.

**Figure 6 nutrients-13-02429-f006:**
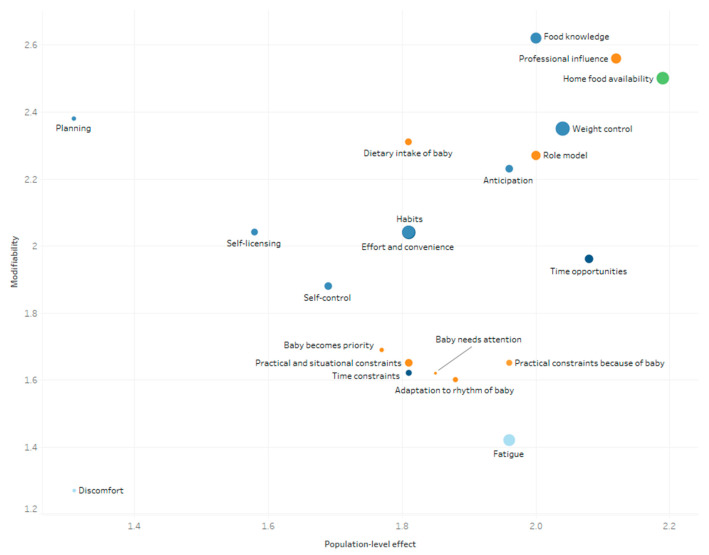
Expert ratings on modifiability, relationship strength, and population-level effect of women’s determinants of changes in eating behavior within the first year postpartum.

**Figure 7 nutrients-13-02429-f007:**
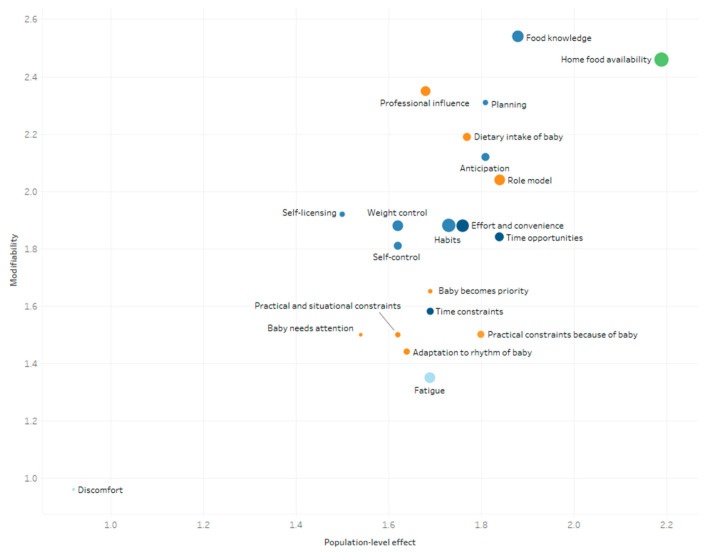
Expert ratings on modifiability, relationship strength, and population-level effect of men’s determinants of changes in eating behavior within the first year postpartum.

**Table 1 nutrients-13-02429-t001:** Characteristics of the study sample of first-time parents.

	Participants Questioned about Determinants during Pregnancy (*N* = 94)		Participants Questioned about Determinants Postpartum (*N* = 85)	
	Women	Men	Women	Men
Sample size (*n*)	57	37	48	37
Age (years)	30.2 ± 2.5	32.7 ± 3.1	32.2 ± 3.1	33.1 ± 3.0
Body mass index (BMI) (kg/m^2^)	24.2 ± 4.9 *	26.0 ± 3.5	23.8 ± 5.3 **	24.5 ± 2.7
Education (% of people with higher or university degree)	91.2	78.3	87.5	78.3
Age of first child when completing questionnaire (months)	19.0 ± 46.0	12.6 ± 2.2	13.4 ± 3.6	13.5 ± 3.7

* *n* = 47 (women expecting their second child not included). ** *n* = 43 (women expecting their second child not included).

**Table 2 nutrients-13-02429-t002:** Characteristics of the study sample of (inter)national experts.

	Female	Male
**Sample size (*n*)**	27	4
**Field of expertise**		
Maternal health *	3	/
Nutrition/physical activity (PA) **	1	2
Public health ***	4	/
Maternal health and nutrition/PA	7	/
Nutrition/PA and public health	3	2
Maternal health and public health	4	/
Maternal health, nutrition/PA and public health	5	/
**Years of expertise**		
<2	3	/
2–5	5	/
5–10	5	/
>10	14	4
**Continent of employment**		
Asia	/	1
Australia	2	/
Europe	21	3
North-America	3	/
South-America	1	/

* Maternal health: gynecology and obstetrics, maternal and child health, maternal obesity, gestational diabetes. ** Nutrition/physical activity: human nutrition and dietetics, physical activity and movement sciences. *** Public health: social and behavioral sciences, public health, epidemiology and health policy.

## Data Availability

The datasets of the current study are available from the corresponding authors upon reasonable request.

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
