# Peer review of "Relative Importance of Determinants of Changes in Eating Behavior during the Transition to Parenthood: Priorities for Future Research and Interventions"

_nutrients, 2021, doi:10.3390/nu13072429_

Round 1

Reviewer 1 Report

I was pleased to revise the manuscript “Relative importance of determinants of changes in eating behavior during the transition to parenthood: priorities for future research and interventions” I find it very interesting and recommend for publication after minor revision.

  1. The introduction is disproportionate to the rest of the work. I suggest adding the results of studies on, for example, obesity among parents of young children.
  2. I suggest adding a separate paragraph for the purpose of the work. It is necessary to add research questions.
  3. I doubt whether such a small population of examined experts entitles the authors to publish data in this field. From my perspective, this is not justified. This is especially true of statistical inference, where it is sometimes based on the answer of one person. I suggest deleting this part of the work and setting it solely around research carried out on the parental population.
  4. With regard to the description of the research procedure, it is necessary to refer to the very low rate of response rate. The authors do not explain why such a low level came from and what steps were taken to prevent it.
  5. Note on Tables 1-2. I am not convinced that all elements should be described in great detail. I suggest shortening the tables, referring to the examples. Such a detailed description unnecessarily lengthens the text and makes it less readable. I also suggest separating the psychological category into two concerning the emotional and cognitive level.
  6. The description of the test results (line 277-289) is sketchy. It has not been described which gender differences are statistically significant.
  7. Figures 2-3 are illegible. Category descriptions are very blurry.
  8. The discussion is a strong point of the paper. However, I suggest changing its shape. I suggest moving the description of biological determinants to the front of the discussion.

Reviewer 2 Report

Thank you for the opportunity to review this interesting manuscript. This paper has a relevant focus for the target population and is of a great value for future research and interventions. The research has a simple and direct design to meet the research questions and is of interest for researchers and clinicians aiming at improvement of eating behaviours during the transition to parenthood.

The Introduction clearly addresses the population with strong literature. It is concise, flows well from section to section, and has a clear thesis statement and aims.

The Methods and Results section are also clear and well organized.

The discussion section is also well organized and reads very well. There is good use of comparison/contrasting literature. The findings are clear and concise. Well done for highlighting some clinical implications of your intervention.

Congratulations for your work, I have no remarks to suggest. I wish you the best for your work. 

Author Response

We thank the reviewer for the very positive feedback on the originally submitted manuscript and for acknowledging the relevance of our work.